# Classification of Thermally Degraded Concrete by Acoustic Resonance Method and Image Analysis via Machine Learning

**DOI:** 10.3390/ma16031010

**Published:** 2023-01-22

**Authors:** Richard Dvořák, Zdeněk Chobola, Iveta Plšková, Rudolf Hela, Lenka Bodnárová

**Affiliations:** Institute of Physics, Faculty of Civil Engineering, Brno University of Technology, Veveří 331/95, 602 00 Brno, Czech Republic

**Keywords:** concrete, high temperatures, nondestructive testing, machine learning, image analysis, Impact-Echo, resonance method

## Abstract

The study of the resistance of plain concrete to high temperatures is a current topic across the field of civil engineering diagnostics. It is a type of damage that affects all components in a complex way, and there are many ways to describe and diagnose this degradation process and the resulting condition of the concrete. With regard to resistance to high temperatures, phenomena such as explosive spalling or partial creep of the material may occur. The resulting condition of thermally degraded concrete can be assessed by a number of destructive and nondestructive methods based on either physical or chemical principles. The aim of this paper is to present a comparison of nondestructive testing of selected concrete mixtures and the subsequent classification of the condition after thermal degradation. In this sense, a classification model based on supervised machine learning principles is proposed, in which the thermal degradation of the selected test specimens are known classes. The whole test set was divided into five mixtures, each with seven temperature classes in 200 °C steps from 200 °C up to 1200 °C. The output of the paper is a comparison of the different settings of the classification model and validation algorithm in relation to the observed parameters and the resulting model accuracy. The classification is done by using parameters obtained by the acoustic NDT Impact-Echo method and image-processing tools.

## 1. Introduction

Concrete is the most widely used construction material worldwide [1,2,3]. This position is due to its rich variability of performance properties, which can be designed for a wide range of construction applications due to the appropriate choice of input materials used. With current concrete 3D printing technologies and the utilization of secondary raw materials on an organic and inorganic basis, the number of possible applications grows each year [4]. Concrete is a solid, noncombustible material with a relatively high thermal capacity (the specific heat capacity *c* of standard plain concrete is 1020 J·kg${}^{-1}\;\cdot \;$K${}^{-1}$). If it is not filled with combustible impurities or fibers, it does not produce smoke at elevated temperatures, but it undergoes physical–mechanical and physical–chemical changes. These changes then affect the material properties of the concrete to varying degrees as the temperature rises and reduce its load-bearing capacity in various ways. The resistance of concrete to this degradation is determined by the composition of the concrete under load, the density and homogeneity of the concrete, and the intensity of the thermal load during exposure to elevated temperature.These basic parameters define the response of the concrete structure to exposure to high temperatures. The first significant changes start to occur at temperatures of 100 °C, when the physically bound water evaporates. At the same time, various types of hydrates may break down, with water beginning to be released from the structure at this temperature. An example is ettringite, which decays in the temperature range 120–200 °C [5]. At the same time, the first part of the decomposition of calcium silica hydrate gels and gypsum gels is beginning to take effect. Despite these initial changes in terms of mechanical properties, there is not such a significant decrease in, for example, compressive strength, tensile strength, or static modulus. A more pronounced decrease in physical–mechanical properties occurs mainly at temperatures of 400 °C [6]. This change is related to the more intensive decomposition of the individual components and also to the increase in water vapor pressure within the concrete structure.

The main course of degradation can be seen in how the individual aggregates and binders react to the elevated temperature. At temperatures above 1000 °C, partial sintering of the mineral components of the aggregate and cement matrix may occur in the concrete. Up to this point, the strength between the aggregate and the cement matrix has been determined by hydraulic bonds, which are replaced at this temperature by the formation of a ceramic bond, accompanied by the formation of new phases, such as Wollastonite. However, when comparing the reaction of different cement composites, this trend does not hold true everywhere. In the published results focused on thermal degradation of cement mortars [7,8,9], a local minimum of physicomechanical properties is reached at 800–900 °C and subsequent sintering continues up to temperatures of 1000–1100 °C. Above this temperature, this type of composite usually enters the melt state and undergoes plastic deformation from its own weight [10]. On the other hand, publications describing thermal degradation of concrete [11] mostly refer to 1000 °C as the local minimum, with partial sintering occurring only above 1100 °C. It is therefore a synergistic effect of many factors, the combined action of which leads to the weakening of the concrete structure and its disintegration. In this area, the thermal degradation process of silicate composites has been described in the works of I. Hager and colleagues [12,13,14], and it is an issue that is widely discussed in the literature.

The current question of how to diagnose the thermal degradation in concrete elements and structures arises with regard to diagnostic approaches. There is no clear answer, and over the past decades many different approaches have been presented. The most accurate approach is to take core bore samples and conduct destructive tests to obtain residual mechanical properties. To be more precise, an X-ray diffraction analysis can discover changes in microstructure, or a visual analysis by both optical and scanning electron microscope can examine the pore structure and cracks in the interfacial transition zone between the aggregate and cement matrix [15]. However, these methods produce a number of parameters that can be related to different processes within the structure of concrete. A regression model can be found that can describe the dependency between results and degradation temperature, and document the changes. For this, a destructive tests needs to be carried out, namely a mechanical residual compressive and tensile strength analysis; in addition, for horizontal elements, such as beams or the ceiling, a static modulus of elasticity needs to be carried out. To supplement these results, an RTG diffraction analysis can discover which phases are present and which are already missing. Mercury porosimetry can describe the size of pores and microcracks within the specimen. All of these methods are expensive, time consuming, and need to be carried out in a laboratory. NDT methods can be done in situ, are fast, and, compared to laboratory methods, relatively cheap.

One of these examples is the Wisconsin bridge diagnosis [16], in which acoustic methods were mentioned as one of the appropriate tools [17]. In the 1990s, the acoustic nondestructive Impact-Echo (IE) method (also known as the resonance method, done by applying mechanical excitations with a hammer (also known as the hammer method)) began to be used to test the thickness of concrete elements. This method is based on the controlled generation of an exciting mechanical wave by means of a mechanical shock and the subsequent sensing of the low-frequency response of the element under test (the method operates in the range of 3 Hz to 20 KHz). In the Czech Republic, this method is hidden under the name of resonance method within the standard CSN 73 1372 [18]. This method has found wide application in the construction industry due to its simplicity, low cost of implementation, and relatively wide range of application possibilities. However, the IE method is dependent on the correct interpretation of the measured data. In practice, it has found application in the measurement of pile lengths, localization of cracks in massive monolithic structures, delamination of bridge bodies, diagnostics of the condition of concrete elements, etc. Due to the simple principle of testing, there are many variations of this method in the form of, for example, the low-frequency pulse echo method or modal analysis.

However, in the past five years a strong trend toward the usage of machine learning (ML) algorithms has arisen [19,20,21]. In 2022, Hatem et al. [21] presented a study of comparison of ultrasound pulse velocity measurement, Schmidt rebound hammer, and compressive strength. In this study, an artificial neural network was used as a predictive tool in which, based on input NDT parameters, a residual compressive strength was predicted. This study focused on a temperature range up to 800 °C, in which the final $R^{2}$ of prediction was in the range of 0.86 to 0.98. This shows the trend of reducing the human factor in the analysis of the unknown specimen. The fact that ML can predict the class of material means that an expert in standard evaluation procedure can check multiple parameters and their range, and make a decision, and that ML can replace this procedure with more parameters, less time, and higher accuracy.

When comparing acoustic methods suitable for in situ testing apart to an IE method, the tested element can be excited also by exciting ultrasonic signal. This methods can be referred as frequency vibroacoustic modulation (VAM) [22] or in some cases nonlinear ultrasonic spectroscopy (NUS) [23]. VAM methods use biharmonic exciting signals, which interact with cracks, delaminations, and other defects within the structure of tested elements. These methods are quite successful in testing close elements and their limitation is mainly the attenuation of exciting signal and the heterogeneity of the tested element. This limitation defines the effective depth, accuracy, and reliability for which these methods can be used, in which higher frequencies are more attenuated than lower frequencies. Without the ML methods, the measured response signals, are quite hard to interpret even for a trained expert, but the ML-based model can find dependencies and increase the overall reliability and accuracy in defect localization or quantitative evaluation of material degradation state. The big advantage of VAM or NUS over IE is the stability of the exciting signal, which has high repeatability; however, both methods are more demanding in terms of hardware than the IE method.

These methods result in response signals, in which a selected feature, such as a dominant frequency or the energy of the signal, is assessed. The usual approach is to use a regression analysis and connect the change in observed features with the change of material properties, such as the level of degradation (represented by the compressive or tensile strength), presence of defects, or different factors. The resulting regression model can then be used to predict the state or presence of the defect. Each observed feature is another dimension, wherein a correlation can lead to different thresholds and criteria, whereas a human observer has low effectiveness and takes a great deal of time. Machine learning algorithms solve this problem, and are able to find the dependencies and criteria much faster and more effectively than a human can. In the presented paper, the acoustic signal and images are reduced to a set of numerical features. For the classification of such a dataset, either machine learning or deep learning algorithms may be used. The main differences are presented on Figure 1 [24].

Deep learning can solve both problems of classification. It can select the features and build a classification model on its own, but it is hard to describe or document why a particular neural network makes such decisions. Moreover, because the DL selects its own features, it is much more demanding on computing power and time needed for feature extraction and training. Machine learning is, on the other hand, dependent on handcrafted feature extraction, for which a function or a program is needed to obtain the training dataset. The model building and training of the ML algorithm is much less demanding in terms of computational power and time, and can be done on standard PCs, handheld devices, or microcontrollers, which enables ML models to be broadly used. This is the main reason why the authors of this paper decided to use ML over the DL approach.

## 2. Materials and Methods

The produced test specimens from each mixtures were divided into several temperature sets, a reference set, and then individual degraded sets. From each set, a set of test bodies was selected to be tested by destructive testing after performing nondestructive testing by using the IE method. For the purpose of this work, in cooperation with Professor Hela, various mixtures were designed, the parameters of which guarantee sufficient diversity of test specimens and therefore establish limit states that will allow future interpolation of measured data. The nondesigned recipes can be divided into two groups. The scheme presented in Table 1 shows which parameters were varied in each recipe.

The first group, which includes mixtures A, B, and C, focuses on the effect of the coarse aggregate used on the results from IE measurements after thermal degradation. The second group includes mixtures D and E and focuses on the effect of different cement types on the IE results after thermal loading. For the composition of each recipe, see Table 2.

These mixtures were designed within the framework of project GAČR No.1602261S. The intention was to design recipes that differ in quality and resulting material properties. Although the influence of the aggregate composition modifies the resulting physical–mechanical properties such as strengths and bulk density, the type of binder used predominantly influences the high temperature response of the composite. All test bodies were stored in a water bath for 28 days after demolding. They were then removed, dried, and left to dry for a week. Before firing, the test bodies were predried at 105 °C for 72 h in a laboratory oven to prevent explosive spalling and unintended destruction of test specimens.

The temperature loading was carried out in the Rhode KE130 furnace, in which the temperature rise was set at 300 °C· s${}^{-1}$. The temperature hold was for 1 h, and the test bodies were then allowed to cool spontaneously to the laboratory temperature of 22 °C. An illustration of the temperature curves is shown in Figure 2.

### 2.1. Resonance Method

The resonance method, also known as the Impact-Echo method, is an acoustic nondestructive method based on the principle of introducing a mechanical shock to the test body and recording the response of the test body to this excitation pulse. It is a common nondestructive method used in defectoscopy in various technical fields. Its particular variant, IE, is adapted for testing building materials, elements, and structures. Mechanical waves can propagate in three forms:longitudinal waves (P-wave),item transverse (shear) waves (S-wave), andsurface waves Raigley waves (R-wave).

These waves move through the material at a speed depending on the acoustic impedance *Z* of the material being measured. If the wave hits the interface of materials with different acoustic impedance, the mechanical energy of the wave is reflected, refracted, or absorbed. For an example, air has an acoustic impedance of 1.275 kg·m${}^{-2}\;\cdot \;$s${}^{-1}$, whereas concrete has an acoustic impedance equal to $10.35\times 10^{6}$ kg·m${}^{-2}\;\cdot \;$s${}^{-1}$. This phenomenon is described by Snell’s law [25]. Due to this phenomenon, mechanical waves are reflected at the interface between the cement composite and the air cavity or surrounding environment. At the same time, mechanical waves interfacing with cracks, steel, and other materials that may be embedded in the concrete mass are affected. As a result, the incidental mechanical wave on the piezoceramic sensor has complex characteristics. The procedure of testing by IE method is illustrated in Figure 3.

The standard process for evaluating this waveform consists of applying a fast Fourier transform (FFT) and converting the measured signal from the time domain to the frequency domain. Then the dominant frequencies that can be assigned to the expected shape frequency are evaluated. In addition to these parameters, however, many other parameters can be assessed on the frequency spectrum that are no longer covered by the standard NDT approach within the IE method. Therefore, the aim of this paper is to verify whether the use of even nonstandard parameters obtained from measured acoustic signals can improve the accuracy in the classification of thermally degraded plain concrete test bodies.

### 2.2. Image Analysis

In addition to the classical NDT method IE, the method of image analysis of the macrotexture of the test bodies was also used [26]. Within the last decade, it has been shown that with advanced image-analysis algorithms and the ever-increasing computing power of even ordinary computers or mobile devices, it is possible to achieve impressive results in defect localization and measurement. Examples include the assessment of cracks in concrete sleepers [27], or the assessment of the degree of thermal degradation based on color change [14]. Image analysis is nowadays quite a broad topic, applying many different approaches. In the context of the presented paper, its simpler form of thresholding and grayscale image binarization was used. For the image acquisition, the designed apparatus produced within project GA22-02098S was used, primarily for the evaluation of cracks in structural composites. Under this project, a Matlab application was designed that operates the camera settings, light controlling via Arduino board, and image acquisition Matlab script. This can be set either to take individual photos or videos. It is also possible to take a time-sequenced photo set of the evolving state of the test specimen. A schematic drawing and photograph of the equipment used is shown in Figure 4.

For the acquisition of photographs, an industrial VCXG-201C.R camera with a resolution of 20 MPx, with a CMOS chip was used. The resolution of the photos is 5472×3648 px. The images were taken with a shutter speed of 6000 μs. The nonilluminated LED strip was powered by a switched 12V DC power supply, which was controlled by an Arduino-type microcontroller by using a mosfet connected to a PC. The camera itself was connected to the PC by using an Ethernet cable. A standard photographic tripod was used to mount the camera with a nonilluminating dome. Between the camera and tripod, a 3D-printed holder was designed.

The light source was a white LED strip glued around the inner perimeter of the shade cover. The shade cover is made of stainless steel, and its inner surface has been anodized to white. This has achieved a satisfactory diffusion of reflected light and uniform illumination of the surface. The body was illuminated for the selected shutter time with a 100-ms advance.

The images were then saved in .png format with a description of the recipe, the temperature at which they were degraded, and from which side of the test body the photo was taken. In most cases, the longitudinal side of the beam perpendicular to the compaction direction was chosen.

The image-binarization method was chosen for image processing. In this method, the color image was converted to grayscale, and a threshold value for binarization was selected. In the case of the presented results, this value was experimentally set to 70 bits. Then, by this condition, all pixels were divided into white pixels that had a value less than 70 bits and black pixels that had a bit value of 71 to 255. A representation of this process is shown in Figure 5.

In this way, the image area was divided into white areas that correspond to cracks, pores, and similar defects and black areas that represent the healthy part of the beam, i.e., the cement matrix, visible aggregate, and other elements of the concrete specimen. It is therefore possible to express what area in a given image is occupied by the observed defects of the surface structure, the surrounding healthy area. The ratio between the number of pixels of the two groups is taken as one of the descriptive variables $I_{R}$.

The summary of extracted parameters includes
average of grayscale image,average blue channel of image,average green channel of image,average red channel of image,contrast value of image,correlation of image,energy of image,entropy of image,kurtosis of image,skewness of image, andratio of white and black pixels.

The feature extraction Matlab script was also designed under project GA22-02098S, and is still under the development. The used method of binarization is dependent on the even distribution of illumination on the examined surface. If some areas are oversaturated, the binarization will falsely highlight these areas instead of defect parts. The ideal lighting would be in the form of the light plane with even intensity at each point, which cannot be achieved in real-world situations. The use of a flash can, on the other hand, oversaturate some light areas and deface the image. In the image processing, LED rings or dark field rings are used, which can solve this problem [28]. An LED light source is much more stable than flashes, with low fluctuations of illumination. Another approach is found in the form of postprocessing by using an adaptive threshold [29], which is able to solve uneven illumination of an examined surface. This solution is good if a localization of the morphological elements where the illumination was not optimal is needed. This postprocessing technique was also tested in the presented paper, but the overall results were not satisfying, because in the case of thermally degraded concrete a change in color is not caused by uneven lighting, but by metamorphological changes in crystalline phases. Apart from simple texture evaluation, it is possible to evaluate individual elements of the macrotexture itself. For example, one paper [30] presents a morphological study of an aggregate and matrix structure of a cross-section of selected concrete specimens. Extracted morphological data were used to simulate the internal volumetric structure of concrete specimens by using the VQ-VAE2 network. This shows that a similar technique could be used to assess the typical morphological defects of thermally degraded concrete. This might be a step for the proposed approach in the presented paper by which to further increase the prediction accuracy of classification of thermally degraded concrete by image-processing tools.

### 2.3. Mechanical and Chemical Analysis

To supplement the NDT measurements, conventional analytic methods were used to describe the degradation from a general point of view. For testing of mechanical properties, a four-point bending test was used. The test procedure was set according to Czech standard CSN EN 12 390-5 [31]. The bending strength and compressive strength together with the static modulus of elasticity are the most commonly used parameters for the evaluation of residual mechanical properties after the thermal loading. In practice, these parameters are evaluated by testing core bore samples removed from thermally degraded structure. These type of tests are usually accompanied by chemical analysis. Because the portlandite mineral decomposes above 600 °C, an X-ray diffraction (RDT) analysis is used to describe the degradation process of individual mineral phases. In the presented paper, RDT was used to document mineral changes in mixture D. For this testing, a device called SAXS Panalytical Empyrean of the AdMas center was used. The RTG analysis was done on a ground cement matrix retrieved from the cross-section of test specimens. Apart from mineral changes in aggregate and cement, pore distribution and size are also affected by elevated temperatures. The pore size rapidly changes in the temperature range of 400–1200 °C [32]. To describe such a change, a mercury porosimetry test is used to describe the distribution of a specific pore volume within the temperature concrete samples. For this type of analysis, a distribution of a specific pore volume based on pore size was evaluated. This means that the result from each test is an array of values, which cannot be directly compared to another array of different samples. One of the ways is to integrate the area below the curve of specific pore volume and compare these areas. The same approach can be found in the paper published by E. Niwa [33]. These tests are presented to describe the behavior of mixture sets from a general point of view. These result are not used in the classification algorithm.

### 2.4. Machine Learning

Machine learning is a subset of artificial intelligence that focuses on developmental systems that learn—or improve their performance—based on the data they work with. Artificial intelligence is a broad term that refers to systems or machines that mimic human intelligence. Machine learning and artificial intelligence are often discussed together, and the terms are sometimes used interchangeably, but they do not mean the same thing. An important distinction is that although all machine learning is artificial intelligence, not all artificial intelligence is machine learning [34]. Basically, machine learning can be divided into three groups:unsupervised learning (clustering);supervised learning (classification); andlearning with feedback (reinforcement learning).

In the presented paper, a supervised classification aproach was used, whereby each test specimen has its class, which describes the temperature at which the specimen was degraded. These classes are 20 °C, 200 °C, 400 °C, 600 °C, 800 °C, 100 °C, and 1200 °C. Each observation is then a set of extracted features from an acoustic signal and a photo of a macrostructure of the test specimen surface. In the presented paper, a set of 197 observations with 41 features is evaluated.

The selected set is then divided into a training set and a test set by the cross-validation method with the k-folds [35]. The input dataset is partitioned into subsets. One subset serves as the test set, and the remaining subsets serve as training sets. The classifier trains a model on the training set and uses the test set to test the accuracy and performance of the model. This process is repeated several times, each time with a different subset forming the training and test sets. In this paper, cross-validation with a fold equal to five was used. An example of such an approach can be illustrated using the flowchart in Figure 6. Apart from cross-validation, another commonly used method is the hold-out validation, in which a dataset is divided into a training and a test group. Usually, the test group comprises 25% and the training group comprises 75%, and each group is picked by random permutation from the whole dataset. This type of validation is suitable for bigger datasets. In training with datasets with less observation and a higher number of observed classes, it can suffer from the fact that some classes are poorly represented, which can generate false-positive results with either abnormally high accuracy or low accuracy. It is important to note that the ratio 75:25 is not a dogma, and some papers utilize different ratios [36].

In the context of research focused on thermally degraded concrete, a consensus can be found across authors on the suitability of nondestructive methods for assessing the degree of thermal degradation of concrete elements and structures. This approach is illustrated in one publication [14] that focused on the use of regression models as a mathematical tool for assessing the degradation rate of plain concrete elements. Nondestructive parameters such as the dominant resonance frequency $f_{L}$, the ultrasonic propagation velocity $v_{UZ}$, or the dynamic modulus of elasticity $E_{dyn}$ are significantly correlated with mechanical properties such as compressive strength $f_{c}$, flexural tensile strength $f_{b}$, and modulus of elasticity $E_{CU}$.

It should be noted, however, that most of these results were obtained from measurements of test bodies and elements. The frequency response of a closed body depends on the shape of the body, the type of anchorage, and the material [37]. Therefore, if these regression models are applied to measurements of a real structure already damaged by fire, the effect of the shape of the structure on the frequency response causes different results, and the regression model obtained from measurement of test specimens cannot be used for measurement of whole structure parts. In contrast, parameters such as the ultrasonic velocity $v_{UZ}$, the acoustic impedance $Z_{UZ}$, or the dynamic modulus of elasticity $E_{dyn}$ can be well used to estimate the residual physical–mechanical properties, which are not influenced by the general shape of the structure [38].

The presented paper also focuses on the evaluation of thermally degraded concrete by using parameters obtained from the IE resonance method and extraction of symptoms from photos of the test body surface. When extracting features from resonant signals, it is possible to focus on several features from both the time domain and the frequency domain. As an example of these flags, Figure 7 shows all the acoustic signal flags measured in the longitudinal testing direction (a) and the main frequency characteristics (b).

The acoustic parameters correlate well for concrete in the range of 20 to 1000 °C when a local minimum is reached for virtually all observed characteristics, both nondestructive and mechanical. However, if the sets degraded at 1200 °C are included, the whole datasets exhibit a nonlinear behavior, in which, due to partial sintering, the observed characteristics increase backward. This leads to possible confusion between sets degraded at 800, 1000 and 1200 °C. Looking at the surface structure, however, it is clear that there is a significant difference between the 1200 °C set and all other temperature groups. This difference can be illustrated by Figure 8.

A similar conclusion can be found in the work of I. Hager [17], wherein the change of color, texture, and morphological elements is associated with the state of thermal degradation of concrete, mortars, and binder pastes. Thus, if the image characteristics from the photo of the surface of the test body and its resonance characteristic are used together, they can separate the groups in the region of 800–1200 °C. Thus, this pair of groups of observed parameters is used to create a test dataset to which various machine learning algorithms can be applied. In this case, multiple classification models can be assessed and compared across the groups. In this case, all observations are divided into seven classes according to the temperature at which the specimens were degraded. A complete list of used extracted features from acoustic signals and images is shown in Table 3.

The presented features were extracted from both IE signals and images of test specimens. To process this data, we used a desktop PC with an Intel i5-10600 3.3 GHz processor, NVIDIA Quadro RTX 4000, 64 GB 2666 MHz DD4 RAM. The average time of feature extraction of signal parameters was 0.05 s, and the average time of image features was 0.69 s. For the model training, we used the Machine Learning Toolbox from Mathworks. This toolbox allows us to pick either cross-validation or a hold-out validation method. Because the amount of observation is under 500, the used dataset of 197 observations is viewed rather as a smaller dataset compared to other examples, such as a Plamer penguin dataset [39]. Hold-out validation is better for bigger datasets, wherein a computational time could be a issue, because cross-validation takes k-times more time for validation. The machine learning toolbox allows a user to test all of the commonly used model presets on the same dataset and is useful for finding the one which has the highest accuracy or is best performing for specific classes within the dataset. With the usage of a parallel computing toolbox, this takes less then two minutes to finish testing all the models. Without the GPU-driven parallel toolbox, the time needed for training all of the models is almost five times longer, but still manageable for a standard office PC. Total consumption of RAM by Matlab during the extraction, training, and testing is 3 GB. The classification of a single specimen then takes less then a second.

Selected features can be influenced by noise as the distortion in input data used for feature extraction and the features itself. The reduction of noise is thus completed by a more reliable feature extraction function, by acoustic filters which reduce the effect of environment on tested body of specimen, or more even illumination or the brushed surface of the test specimen. Another form of noise can be the wrong placement of class labels, in which a sample specimen which degraded at 800 °C was by human error labeled as 400 °C. Generally speaking, the noise influence has a bigger impact in deep neural networks, which are usually used on vast datasets where it is not possible to maintain the stability of the data input. For example, speech-recognition algorithms are usually trained on thousands of records that contain a natural form of language, and the presence of noise is much higher [40]. In the presented dataset, the specimens were prepared under laboratory conditions, and the controlled degradation was carried out precisely for each test specimen. However, the reliability of the trained model can be tested by implementing artificial noise to the data, which can show under what level of noise the model is still stable and reliable [41]. This type of testing was not done in the paper, but will be carried out as a next step, because it has a great value for transferring this diagnostic approach to the industry. Under the real condition in situ, the enviroment has a much bigger impact on the acoustic noise, lightning, and presence of water or other substances.

## 3. Results

In the context of the nondestructive diagnoses of the thermal damage rate, the presented paper is based on the NDT resonance method and its high correlation with conventional destructive methods. This dependence is shown in Figure 9. The dominant resonance frequency in the longitudinal direction $f_{L}$ in Hz is shown on the *y* axis and the flexural tensile strength $f_{B}$ in MPa is shown on the *x* axis. All measured mixtures A–E are shown, and the temperature sets 200–1200 °C are marked. The correlation value is greater than 0.95, so that the resonance frequency value can be used to express the change in tensile strength in the case of the test bodies. However, there are several problems:resonant frequency is a shape-dependent quantity that is influenced by the excitation and mounting of the test fixture,the temperature sets overlap to some extent, especially in the region of 800–1200 ${}^{\circ }$C, and it is almost impossible to safely distinguish the temperature group by using NDT analyses alone, andthe set degraded at 1200 °C corresponds to a different trend, which is due to partial sintering of the concrete, where the material starts to behave like ceramic.

For these reasons, it can be stated that the use of the resonance method as a diagnostic tool to distinguish the degree of thermal degradation can only be reliably used up to 800 °C, on test fixtures of known size, mounting, and excitation.

The significant change between 1000 and 1200 °C can be described by X-ray diffraction analysis, the results of which are shown in Figure 10. The reflection intensity is shown on the *y* axis, and the reflection angle of 2θ is shown on the *x* axis. The colours are used to distinguish between the different temperature sets of the mixture D. From these results, the representation of the individual crystalline components and their change during loading can be discerned. The results show a gradual decrease of portlandite, which is no longer present in the samples after 600 °C; on the other hand, at 1200 °C, the ceramic phase Wolastonite is formed, which is the main cause of the increase in strength and resonance frequencies. It is also the reason for the significant change in the surface texture of the sample in terms of morphology and color change.

The change in macrotexture is one of the NDT indicators that can be obtained relatively easily when evaluating both concrete test bodies and whole structures, which has already been published e.g., in [17]. One of the objectives of this paper is to improve the general reliability of thermal damage classification of concrete test bodies for the whole temperature range 200–1200 °C by using additional parameters obtained by NDT methods. Thus, it is necessary to use not a bivariate regression model but a multivariate evaluation. Machine learning algorithms can be used very successfully for these tasks. The parameters of the measured signals in the longitudinal direction of the test bodies $f_{L}$ were used to create this model. For all signals, the change in characteristics clearly shows a nonlinear dependence on the degree of thermal degradation with a local minimum at 1000 °C. The temperature set of 1000 and 1200 °C overlaps with the temperature set of 800 and partially for the set of 600 °C. An example of this overlap is the rise time $S_{RT}$ parameter shown in Figure 11. This is a representation of the dependence of the dominant resonance frequency $f_{L}$ in Hz on the value of the rise time $S_{RT}$ in s. The individual temperature sets are shown in color. The overlap of the temperature sets can be seen here, starting from the reference sets.

Because these are laboratory-measured specimens, measured under optimum conditions, it is to be expected that in the case of real structure measurements this overlap will occur to a much greater extent and the analytic value of acoustic NDT measurements alone may not be sufficiently accurate. It is therefore desirable to supplement these NDT measurements with additional parameters that allow us to distinguish between the different sets. In this sense, the tests carried out so far and the scientific results are rather focused on the hypothetical use of the IE method in the diagnosis of thermally damaged structures. Examples include a diagnostic survey of a bridge [16], or laboratory testing of the concrete bodies of precast bridge elements [42]. These examples either focus on a hypothetical application of acoustic methods or online monitoring of an ongoing fire test.

From the point of view of in situ measurements, it is therefore a good idea to supplement the IE measurements with an additional variable that will have a linear change as a function of the rate of thermal degradation. An example of such a dependence could be the integrated pore size area from a mercury porosimetry test. An example of this integration is shown in Figure 12, where the *x*-axis shows the increasing temperature and the *y*-axis shows just the integrated pore size area.

This value is almost identical in the range 0–600 °C, but from a stress temperature of 800 °C there is a sharp increase in the integrated pore size value $V_{P}$. At 1000 and 1200 °C, the relative differences are then greatest, and it can be said that this analytic method can be used reasonably well to distinguish temperature groups that are otherwise difficult to distinguish for the NDT IE method or by destructive tests. The mercury porosimetry method is, however, demanding in the preparation of the test sample. A regular sample of dimensions 10×10×10 mm is to be obtained if possible after the test is contaminated with mercury. This method cannot be used in situ either. On the other hand, the macrotexture of the observed material is clearly visible, and based on the publication [17] it is known that there is a dependence between the color, hue and brightness of the concrete structure on the degradation temperature. Taking a photograph of the concrete surface affected by elevated temperature at the test site by using the resonance method is a relatively easy task that does not require expensive equipment and can be performed very well under in situ conditions. If a surface photograph of the test bodies is used, it can subjected to image analysis and evaluation of the parameters such as color, hue, and other parameters. An example of this basic comparison and the individual steps is shown in Figure 13 and Figure 14. The analysis consists of taking a photograph with a stable light source at sufficient resolution. Each RGB photograph is a three-dimensional vector, in which the individual layers contain intensities in the range 0–255 in individual shades of red, blue, and green. By composing all the channels, a color image is produced. For color photographs, it is worth mentioning that they work in the visible spectrum, and it does not precisely represent the color of the material depending on the reflected wavelength, which can be analysed, for example, with spectrophotometers. This analytical test is used to analyse the color of different materials and substances in relation to the reflectance of specific wavelengths of radiation [43]. From this point of view, image analysis is simpler and therefore cannot capture spectral information in, for example, the near-infrared or ultraviolet spectrum or the intensity representation of reflected or transmitted light at a particular wavelength. However, image analysis is able to provide additional valuable information that cannot be obtained in any other way. Specifically, in this paper, an image was created from each photograph taken and analyzed by using the binarization method. Then, all the integral white areas corresponding to the darkest areas in the image were separated, which in the case of thermally degraded concrete corresponds to cracks, pores, and significantly degraded parts of the aggregate that have substantially changed their hue.

The effect of using the parameters from the image analysis of the surface photographs of the thermally degraded concrete bodies can be seen in the distinction between the set degraded to 20 and 1200 °C, as documented by the correlation diagram in Figure 15. Here, the selected main parameters extracted from the photos, namely the average value of the blue channel, the skewness of the image bits in grayscale, the photo energy, and the ratio between the number of white and black pixels from the binary image, are compared in the correlation. This comparison shows the separation of the test set degraded at 1200, 1000 and 800 °C. This separation is most evident when comparing the blue channel with virtually all other parameters. This separation is most evident in the comparison of the image energy and the ratio of white to dark pixels. The remaining 20–600 °C test sets are already mixed in most cases in one cloud of measured points.

However, this comparison is sufficient to differentiate the individual test sets, as can be seen in Figure 16, where the *x*-axis shows the dominant frequency $F_{L}$, the *y*-axis the signal attenuation, and the *z*-axis the average value of the green channel of the macrotexture photo. Here, one can clearly distinguish all the test sets that occupy a specific place in the NDT measured parameters. The machine learning algorithm can be used to verify the success of the proposed evaluation system. To apply this method, the measured data needs to be transformed into a dataset structure, in which each row represents just one observation consisting of columns containing parameters (features) from the IE method and image analysis. The last column contains information about the class of observation, in our case the temperature to which the beam was degraded. The evaluation algorithm is based on the principle of cross-validation, whereby the whole dataset is split into a 75:25% ratio, where 75% serves as the learning set and 25% serves for validation without knowing the actual class of the observation. Both parts are chosen by using random permutation so that all unique class types are equally represented.

A classification model for distinguishing the degree of thermal degradation by using NDT parameters obtained by the IE method and image analysis was designed by using the Machine Learning Toolbox of Matlab. This toolbox offers a range of different classification algorithms presets and all of them can be applied on the same dataset and their efficiency, success rate, and model accuracy can be compared with each other. Thus, from this analysis, the ensemble bag tree (BT) type was used as the final classification model with the highest accuracy. EBT combines a set of trained weak-learner models, and the data on which these learners were trained. It can predict the ensemble response to new data by aggregating predictions from weak learners [44]. The bagging technique consists of three steps.
The original data is separated into subset data. The columns and rows of original data are divided into subsets of data.Classifiers are built on each subset of data. The same or different classifiers are created for subset data.The majority vote is used for choosing the best classifier from all classifiers [45].

It stores the data used for training, can compute replacement predictions, and can continue training if necessary. The resulting classification model provides a prediction based on the input parameters from the IE method and image analysis of what class of temperature degradation is involved. The success rate can be represented by using the confusion matrix shown in Figure 17. Here, the matrix compares the success rate of the classification of classes where the correctly predicted classes lie on the diagonal. If an observation was not correctly classified, it can be expressed with the other class with which the proposed model confuses this observation.

The total time needed for the training of the ensemble bagged tree model is 2.6 s, and the accuracy is 85%. With optimization, the training time is 51 s with a resulting accuracy 87%. The hyperaparametrs for the optimized model were altered by using the optimization tool and are presented at Table 4. During the optimization procedure, the program changes the hyperparameters in a given range and we can observe the estimated and observed minimal classification error. A similar approach can be seen in different, related machine learning tool applications [46].

The resulting accuracy of this model is 87.31%, where there is only confusion between adjacent temperature sets. The test set degraded at 1200 °C is so unique that there was no confusion with any other test set during classification. The greatest confusion occurs between 400 and 800 °C, where the successful classification rate for 600 °C is 76.3%.

The resulting decision tree generated by the optimized bagged tree classifier is presented at Figure 18. It can be seen that the first parameter is a BW ratio $I_{R}$ of an image and subsequent conditions are mainly by dominant frequency. From all of the parameters that were given as an input dataset for model training, these parameters have the highest impact on successful classification. It is important to note that the training is possible with the usage of principal component analysis [47], which can reduce the dimensionality to a couple of principal components. This approach was also tested in the training of the presented dataset, but the result was not satisfactory, mainly because the PCA combines all of the variables into principal components, and the image feature information is suppressed. Other approaches are seen in deciding what parameters are most important for classification as can be seen in [48] where the mean squared error (MSE) is observed at the different regularization parameter λ. In the presented paper, the selected approach picked the most significant features by the model itself.

If the obtained features from the image analysis in the classification would not been used, the success rate of the classification model of 73% would be 14% lower, with the classification accuracy of the temperature sets of 600, 800, 1000, and 1200 °C being only 60%. From this difference, the positive effect of using image analysis in the classification of thermal damage can be clearly documented. Better results could have been achieved by treating the surface of the thermally degraded beams, for example by removing the surface layer with a wire brush or by using a grinder; however, the aim of the experiment was to validate the NDT approach, which can be performed repeatedly on many test pieces or structural elements without any surface treatment.

The influence of the mixture used on this classification success rate does not have a significant effect, as can be seen by comparing the success rates between the different recipes in Table 5. This table compares the success rates of the proposed classification model for each tested recipe. The lowest value is achieved by recipe C with a success rate of 80.48% and the highest value is achieved by recipe B with a success rate of 88.88%. Thus, this is a variance of 8%, which can be attributed mainly to the atypical composition of recipe C, wherein the coarse fraction was completely absent. This recipe generally has a much larger variance in terms of measured parameters, and therefore its results can be expected to extend into parameter regions typical of neighboring temperature sets. Statistical parameters such as mean, minimum, and maximum of each feature is presented at Table 6.

All of the presented extracted features are stored on Web Page figshare, in Appendix A. The dataset has 41 columns for either signal or image features, one column for temperature class and one column for mixture.

## 4. Conclusions

In the presented paper, a procedure for the evaluation of thermally degraded concrete by using the NDT Impact-Echo method and image analysis of the surface of degraded concrete beams was proposed. Five different mixtures were measured in their entirety, and the effect of varying the proportion of coarse aggregate and the binder used was investigated. The proposed evaluation procedure is based on the extraction of characteristic features from the measured acoustic signals obtained by the IE method and features obtained from the photographs of the test mixtures by the image-analysis method. The classification procedure was evaluated by using a machine learning method applying cross-validation with a 75:25 split with five folds of the entire dataset. The proposed classification model achieves a classification success rate of 87%, where confusion between temperature sets 20–1200 °C occurs only between adjacent temperature sets. If the same model was used only on data obtained from the IE method, the accuracy would have at most a 73% success rate with a high degree of confusion across the temperature range of 600–1200 °C. Classification accuracy by using only the parameters from the image analysis would achieve 69% with a high confusion rate across the entire temperature range. Thus, it can be stated that it is the combination of the parameters from the IE method and image analysis that allows for higher reliability in the classification of thermal damage in plain concrete test bodies by using only nondestructively obtained parameters. At the same time, the proposed procedure for validating the accuracy of the classification model by using cross-validation appears to be a suitable tool for validating the accuracy of similar classification tasks due to its simplicity and versatility. It is important to note that the presented data were prepared under laboratory conditions, and the influence of noise is very limited. The next step would be implementing artificial noise factors to both signal and image characteristics, and validating the accuracy and repeatability of noise in the data at different levels. For the increasing classification accuracy only for image features, a deep learning algorithm with morphological segmentation of characteristic defects of the aggregate and cement matrix would be beneficial.

## Figures and Tables

**Figure 1 materials-16-01010-f001:**
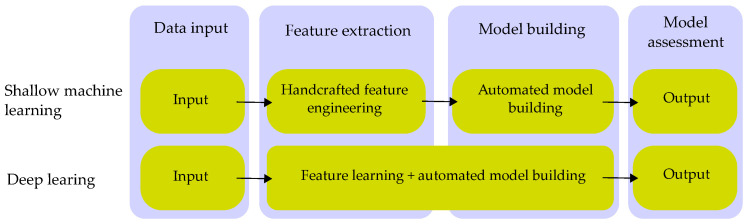
Differences between ML and DL approach [24].

**Figure 2 materials-16-01010-f002:**
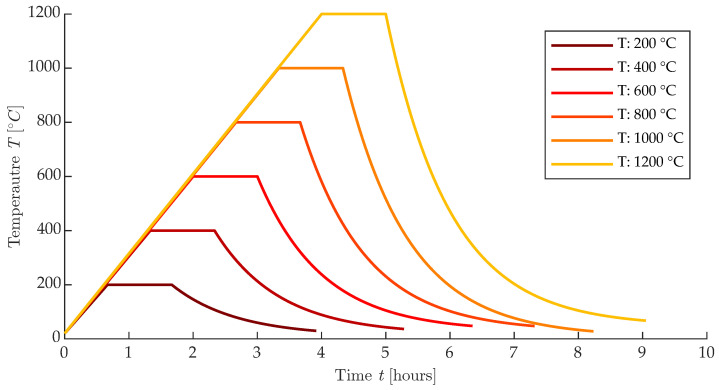
Illustration of temperature curves for all degraded groups.

**Figure 3 materials-16-01010-f003:**
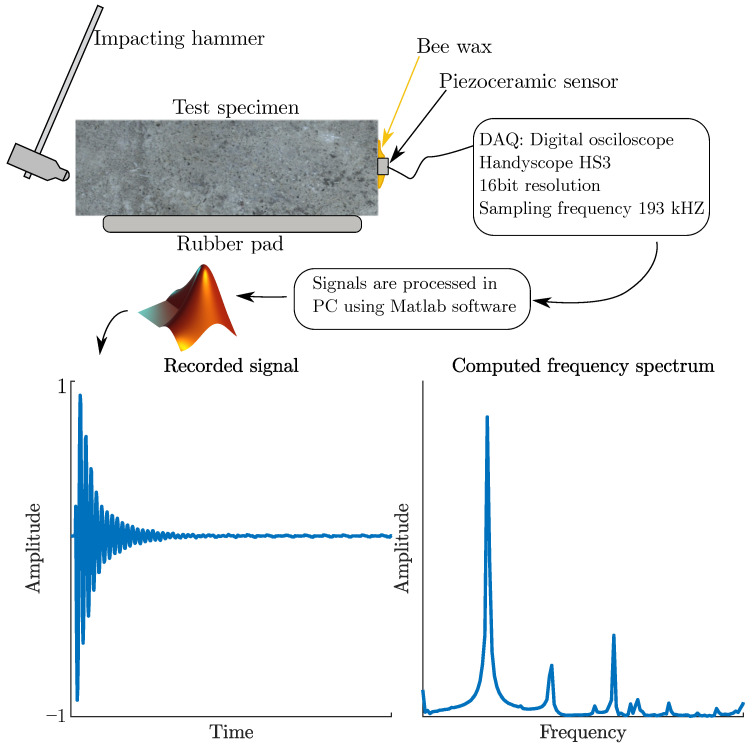
Illustration of measurement process by IE method.

**Figure 4 materials-16-01010-f004:**
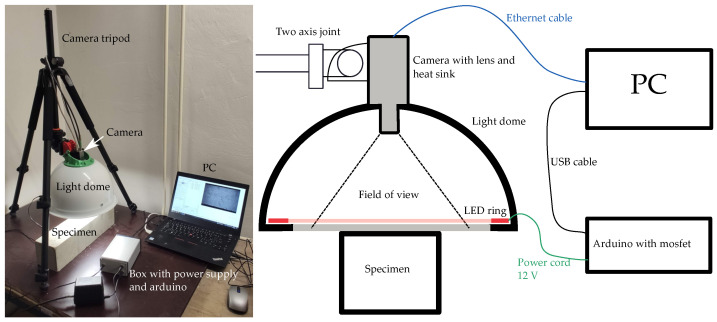
Scheme of camera setting used for image acquisition.

**Figure 5 materials-16-01010-f005:**
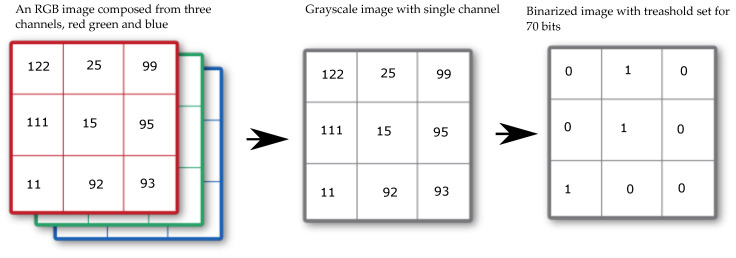
Process of binarization of an RGB image.

**Figure 6 materials-16-01010-f006:**
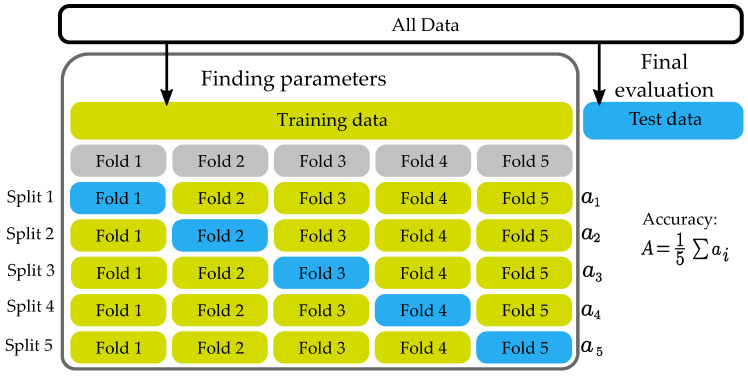
Five fold cross-validation.

**Figure 7 materials-16-01010-f007:**
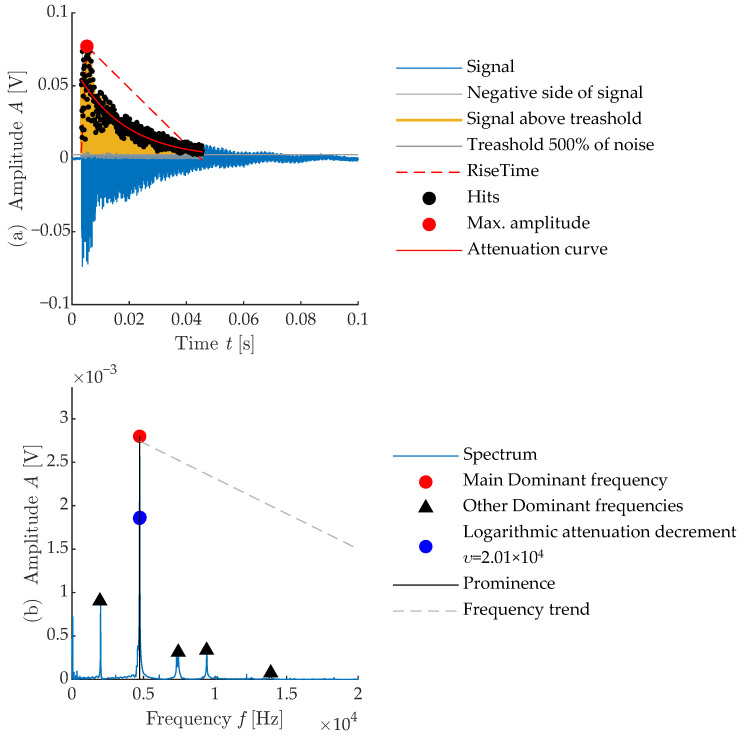
Visualizaton of feature extraction done on impulse signal.

**Figure 8 materials-16-01010-f008:**
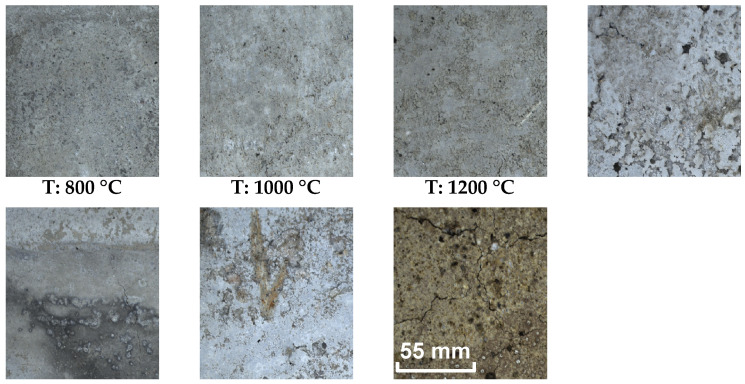
Comparision of different color and texture of concrete specimens from 20 to 1200 °C.

**Figure 9 materials-16-01010-f009:**
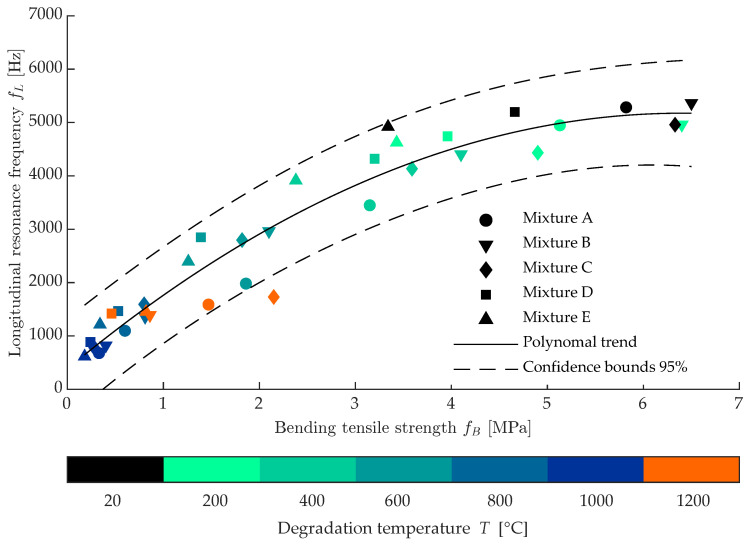
Correlation comparision of longitudinal frequency $F_{L}$ and bending tensile strength $f_{B}$.

**Figure 10 materials-16-01010-f010:**
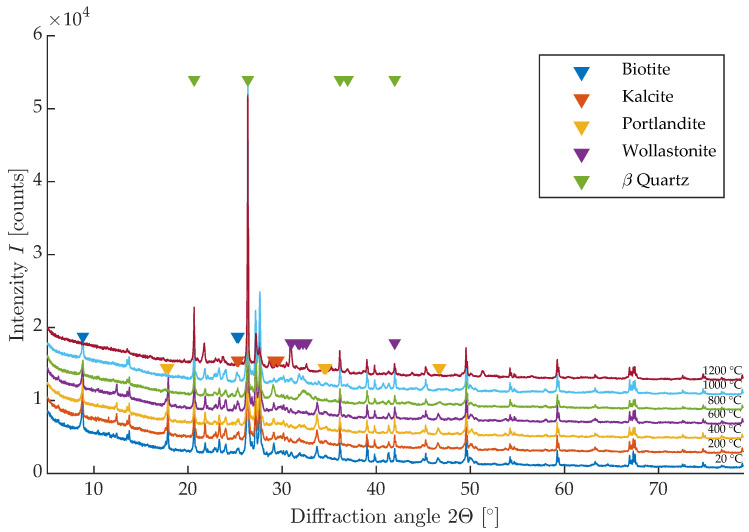
RTG diffraction analysis of tested mixture D.

**Figure 11 materials-16-01010-f011:**
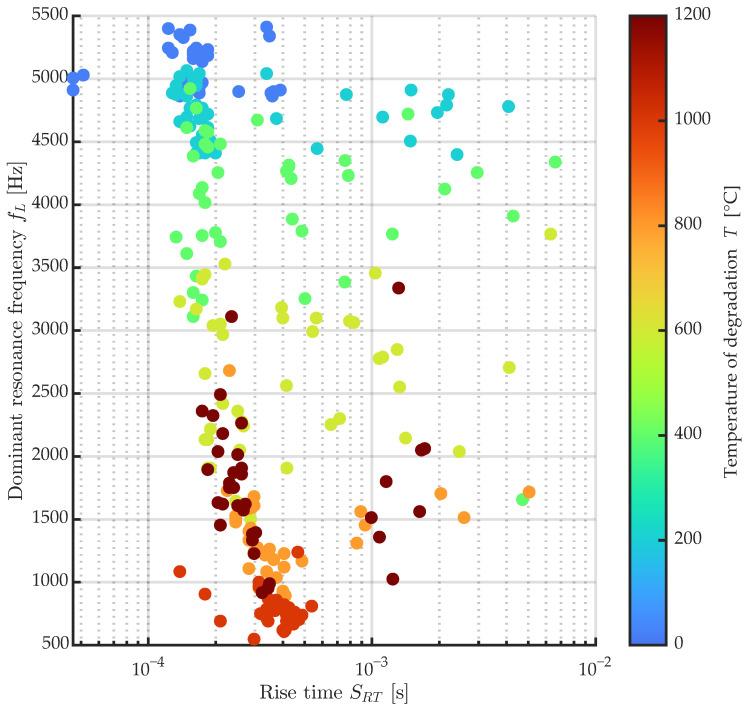
Dependency of parameter longitudinal resonance frequency $f_{L}$ on a rise time RT of signals of all mixtures across the temperature sets 20–1200 °C.

**Figure 12 materials-16-01010-f012:**
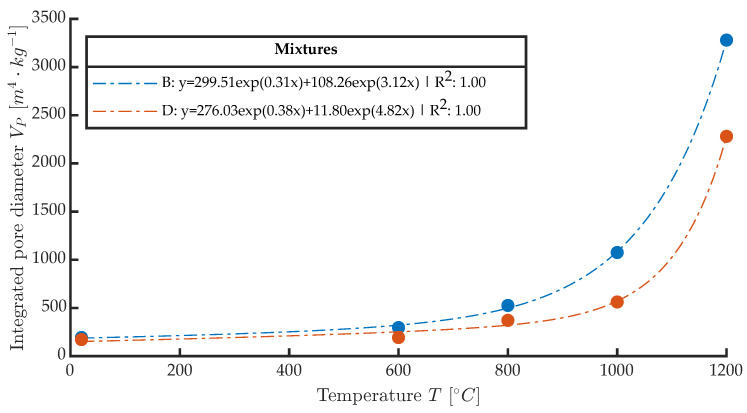
Integrated pore size of mixtures B and D from mercury porosimetry test.

**Figure 13 materials-16-01010-f013:**
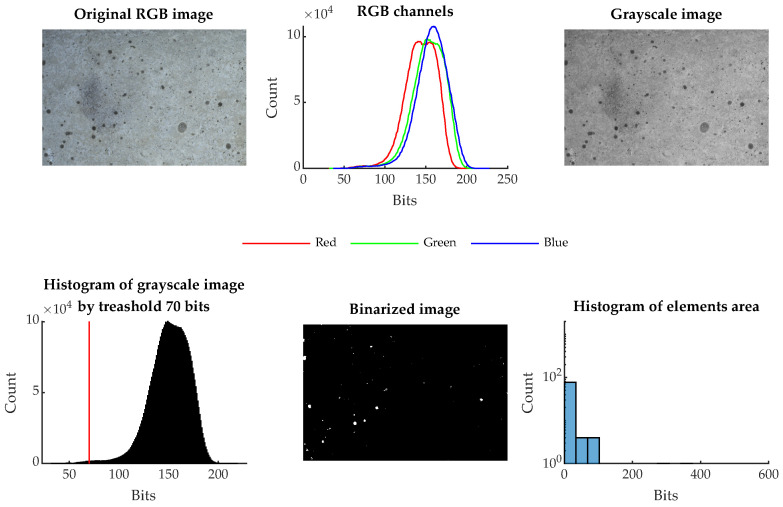
The process of image feature BW ratio $I_{R}$ for specimen degraded at 20 °C.

**Figure 14 materials-16-01010-f014:**
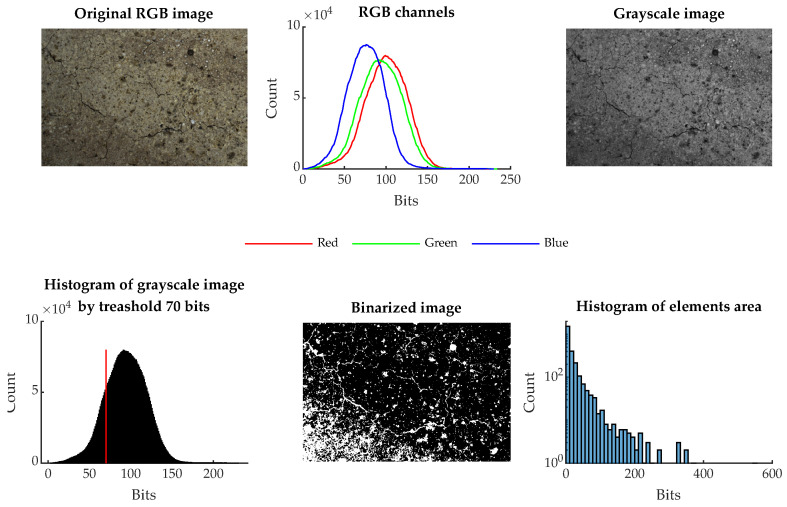
The process of image feature BW ratio $I_{R}$ for specimen degraded at 1200 °C.

**Figure 15 materials-16-01010-f015:**
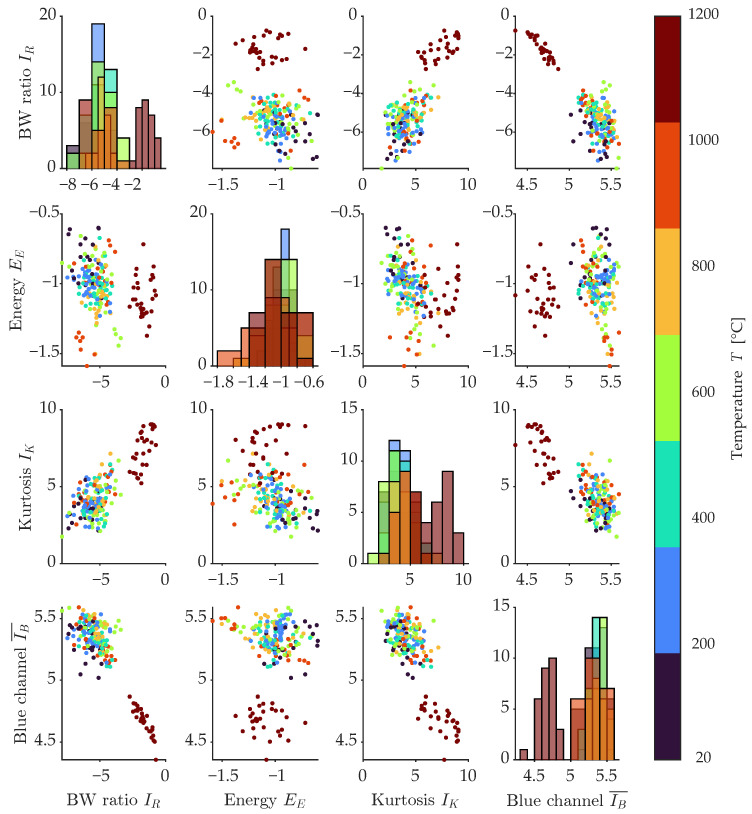
Correlation diagram of selected variables: $R_{BW}$, $E_{I}$, $K_{I}$, $\sum G_{channel}$.

**Figure 16 materials-16-01010-f016:**
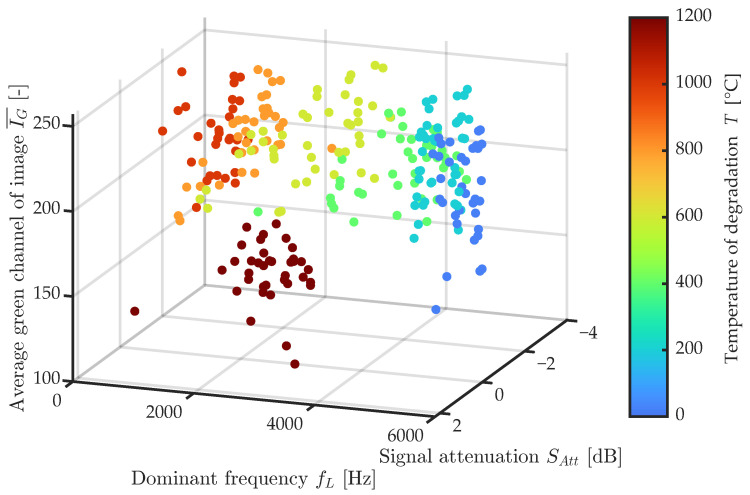
Comparison of dependency between green channel average value, dominant frequency $f_{L}$, and signal attenuation.

**Figure 17 materials-16-01010-f017:**
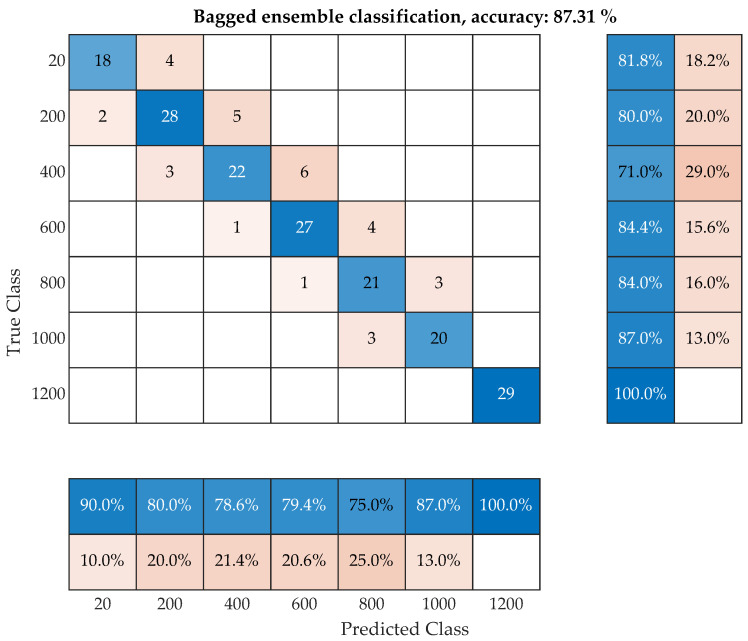
Confusion matrix of proposed machine learning model by using support vector machine algorithm.

**Figure 18 materials-16-01010-f018:**
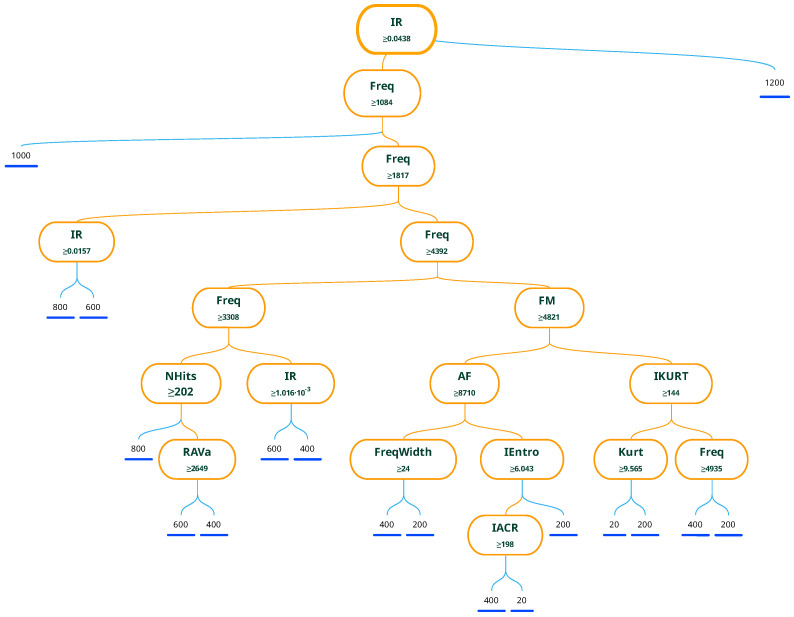
Decision tree of optimized bagged tree model.

**Table 1 materials-16-01010-t001:** Design of manufactured mixtures.

Change of Binder	Change of Coarse Aggregate		
	0/4,8/16	0/4,8/16,11/22	0/4,4/8
CEM II/A-S 42.5 N		E	
CEM I 52.5 R		D	
CEM I 42.5 R	A	B	C

**Table 2 materials-16-01010-t002:** Composition of designed mixtures.

Compound	Weight per 1 m^3^ [kg]				
	A	B	C	D	E
CEM I 42.5 R	345	345	345	-	-
CEM II/A-S 42.5 N	-	-	-	345	-
CEM I 52.5 R	-	-	-	-	345
Fine aggregate Žabčice 0/4 mm	848	813	813	934	934
Coarse aggregate Olbramovice 4/8 mm	-	-	1010	-	-
Coarse aggregate Olbramovice 8/16 mm	980	521	-	355	391
Coarse aggregate Olbramovice 11/22 mm	-	391	-	355	391
Admixture SikaViscocrete 2030	2.8	2.5	3.1	3.1	3.1
Water	160	176	176	155	162

**Table 3 materials-16-01010-t003:** List of all used features from signals and images.

ID	Name	Description	ID	Name	Description
1	$S_{A,d}$	Rise slope	23	$I_{E}$	Energy of image
2	$S_{A,RA}$	RA asymptote	24	$I_{S}$	Entropy of image
3	$S_{Att,R2}$	$R^{2}$ of signal attenuation	25	$I_{K}$	Kurtosis of image
4	$S_{Att,SSE}$	SSE of attenuation curve	26	$I_{Skew}$	Skewness of image
5	$\bar{I_{RGB}}$	Average of grayscale image	27	$S_{IF}$	Impulse factor of signal
6	$\bar{I_{B}}$	Average blue channel of image	28	$S_{K}$	Kurtosis of signal
7	$\bar{I_{G}}$	Average green channel of image	29	$S_{A,max}$	Maximum amplitude of signal
8	$\bar{I_{R}}$	Average red channel of image	30	$S_{c}$	Signal peaks count
9	$\bar{f_{L}}$	Mean of dominant frequencies	31	$f_{Asym}$	α of frequency trend peaks
10	$I_{contrast}$	Contrast value of image	32	$f_{\beta }$	β of frequency asymptote
11	$I_{cor}$	Correlation of image	33	$S_{RMS}$	Root mean square of signal
12	$S_{CF}$	Crest factor of signal	34	$I_{R}$	Ratio of white and black pixels
13	$f_{\vartheta }$	Decadic attenatuion coefficient	35	$S_{RA}$	Rise angle of signal
14	$S_{dur}$	Duration of signal above threshold value	36	$S_{RE}$	Energy of signal in RA region
15	$S_{E}$	Energy of signal above the threshold	37	$S_{RT}$	Duration of rise angle range
16	$f_{L}$	Frequency with maximum amplitude	38	$S_{SNR}$	Signal-to-noise ratio
17	$f_{A,max}$	Amplitude of dominant frequency peak	39	$S_{Att}$	Signal attenuation
18	$\bar{f_{AF}}$	Average frequency of signal	40	$S_{Skew}$	Skewness of signal
19	$f_{L,c}$	Count of dominant frequency peaks	41	$S_{SV}$	Absolute voltage range of signal
20	$\Sigma f_{L}$	Sum of dominant peaks	42	$S_{THD}$	Total harmonic distortion of signal
21	$f_{L,w}$	Peak width	43	$S_{Trsh}$	Threshold 150% of noise of signal
22	$I_{H}$	Homogenity of image			

**Table 4 materials-16-01010-t004:** Hyperparameters of the trained ensemble bag tree model.

Hyperparameters	Ranges	Optimized Values
Ensemble method	Bag, AdaBoost	Bag
Learner type	Decision tree	Decision tree
Number of learners	10–500	31
Maximum number of splits	1–196	187
Number of predictors to sample	1–43	34

**Table 5 materials-16-01010-t005:** Classification accuracy among all tested mixtures.

Accuracy [%]	Mixture				
	A	B	C	D	E
	85.29	88.88	80.48	85.71	88.57

**Table 6 materials-16-01010-t006:** Comparision of all extracted features.

ID	Parameter	Mean	Min	Max	ID	Parameter	Mean	Min	Max
1	$S_{A,d}$	0.070	−1.522	4.905	23	$I_{E}$	0.366	0.204	0.549
2	$S_{A,RA}$	2.354	0.075	14.750	24	$I_{S}$	6.353	5.681	7.016
3	$S_{Att,R2}$	0.669	0.041	0.987	25	$I_{K}$	567.091	5.755	8531.601
4	$S_{Att,SSE}$	0.296	0.002	3.736	26	$I_{Skew}$	13.921	1.732	92.037
5	$\bar{I_{RGB}}$	194.305	101.357	255.089	27	$S_{IF}$	49.929	10.637	160.686
6	$\bar{I_{B}}$	196.874	77.965	268.100	28	$S_{K}$	10.294	2.899	32.426
7	$\bar{I_{G}}$	197.156	106.299	257.182	29	$S_{A,max}$	0.200	0.029	0.858
8	$\bar{I_{R}}$	188.885	119.179	241.034	30	$S_{c}$	550	103	1334
9	$\bar{f_{L}}$	2982.401	619.879	5412.019	31	$f_{Asym}$	0.584	−$3.65\times 10^{-8}$	1
10	$I_{contrast}$	0.078	0.044	0.145	32	$f_{\beta }$	$6.66\times 10^{-3}$	−$8.7\times 10^{-3}$	$1.84\times 10^{-3}$
11	$I_{cor}$	0.921	0.838	0.957	33	$S_{RMS}$	0.040	0.009	0.177
12	$S_{CF}$	6.062	2.957	10.450	34	$I_{R}$	$3.83\times 10^{-3}$	$3.78\times 10^{-3}$	$4.76\times 10^{-3}$
13	$f_{\vartheta }$	$1.03\times 10^{9}$	$1.39\times 10^{9}$	$1.79\times 10^{9}$	35	$S_{RA}$	839.132	19.643	6706.851
14	$S_{dur}$	0.054	0.002	0.158	36	$S_{RE}$	0.242	0.035	0.991
15	$S_{E}$	$2.96\times 10^{-3}$	$4.72\times 10^{-3}$	$7.68\times 10^{-3}$	37	$S_{RT}$	$5.87\times 10^{-3}$	$4.61\times 10^{-3}$	$6.60\times 10^{-3}$
16	$f_{L}$	2938.715	607.958	5412.019	38	$S_{SNR}$	23.959	8.687	39.032
17	$f_{A,max}$	0.000	0.000	0.000	39	$S_{Att}$	−1.830	−3.780	2.228
18	$\bar{f_{AF}}$	$1.19\times 10^{5}$	$5.98\times 10^{5}$	$5.70\times 10^{5}$	40	$S_{Skew}$	−0.470	−3.099	0.594
19	$f_{L,c}$	1.474	1.000	4.000	41	$S_{SV}$	0.449	0.070	1.858
20	$\Sigma f_{L}$	0.000	0.000	0.000	42	$S_{THD}$	−13.211	−56.395	2.272
21	$f_{L,w}$	26.045	17.216	64.537	43	$S_{Trsh}$	0.007	0.001	0.023
22	$I_{H}$	0.961	0.927	0.978					

## Data Availability

Extracted features used for training of machine learning algorithm are stored on figshare under https://figshare.com/articles/dataset/Extracted_features_from_acoustic_Impact-Echo_method_and_image_processing_procedure_of_thermally_degraded_concrete_specimens/21702149, accessed on 9 December 2022.

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
