# Peer review of "Classification of Thermally Degraded Concrete by Acoustic Resonance Method and Image Analysis via Machine Learning"

_materials, 2023, doi:10.3390/ma16031010_

Round 1
Reviewer 1 Report
Check that the style of writing is in the third person throughout. Don’t use ‘we’.
Machine Learning is a broad term, and using it in the title may not lure a potential reader into noticing. I would recommend using your model (algorithm) name in the title.
Why did you use ML algorithms over a deep learning approach?
Comment on the explainability (white-box approach) of the model proposed. A white-box model may be provided for understanding the judgment, or you may suggest it in future scope. You can refer following articles to lure potential readers: A White-Box SVM Framework and its Swarm-Based Optimization for Supervision of Toothed Milling Cutter through Characterization of Spindle Vibrations.
Was the algorithm trained using standard hyperparameters, or were they altered? You may refer to the recent article, Augmentation of decision tree model through hyper-parameters tuning for monitoring of cutting tool faults based on vibration signatures.
Discuss the effect of kernel functions and regularization factor ‘C’ of SVM while training your data.
Hyperparameters of the algorithm must be summarised in tabular form.
Comment on computational time and complexity in the training of the algorithm.
Was the data normalized/ standardized?
How to ensure the robustness of the model in a highly noisy environment?
Additionally, results can be provided considering different holdout % and holdout validation approaches. You may refer to this article to understand the holdout validation approach. https://doi.org/10.1115/1.4051696
How to deal with the diversity between the data distributions of present and future moments? ML-based algorithms can only resolve the regression/classification issues within the same data distributions. What would be the key steps of generalizing to unknown moments in predicting various parameters?
The authors do not mention the availability of this platform or framework for other practitioners, which is mandatory for this type of application. Add a comparison table at the end of the results section.
Author Response
Reviewer 1:
Dear Reviewer,
I am deeply thankful for your time. Your review is a thorough, and it helped me a lot to think about the paper and all of the ML related topics. During my revision I have found many different paths, which I can take in the next research, and Your questions and advices were most helpful. Here are my responses to Your review:
Check that the style of writing is in the third person throughout. Don’t use ‘we’.
I have check it in the whole paper, it should be correct now.
Machine Learning is a broad term, and using it in the title may not lure a potential reader into noticing. I would recommend using your model (algorithm) name in the title.
In the field of civil engineering the machine learning approaches are being used relatively for short time, and lot of my collegues view it still as something ‘not easy to understand’. The possible different title would be ‘Classification of thermally degraded concrete by acoustic resonance method and image analysis via Bag trees classification model’, but I am afraid that it could be misleading. Machine learning is a broad term, but easily recognisable and almost all researchers heard about it somewhere. If you think, it will be beneficial for my paper, I will ask the editor and I will change it.
Why did you use ML algorithms over a deep learning approach?
Mainly because the simplicity, speed and demands for hardware. I’ve used deep learning for machine vision classification and morphological classification, for this is DL perfect, but for simple numerical matrices classification learners can work better in my honest opinion. I have tried to explain it in the paper.
Comment on the explainability (white-box approach) of the model proposed. A white-box model may be provided for understanding the judgment, or you may suggest it in future scope. You can refer following articles to lure potential readers: A White-Box SVM Framework and its Swarm-Based Optimization for Supervision of Toothed Milling Cutter through Characterization of Spindle Vibrations.
I had wrong title in confusion matrix, I used bagged trees instead of SVM. Utilization of ML algorithm in industry is based on its accuracy, stability, and credibility. The diagnostic experts in civil engineering are often closely working with building statics, which believe only in reliable methods, and real proofs. From this point of view, it is more likely to use a white box approach rather than a black box, which is exactly what we are trying to achieve with our paper.
Was the algorithm trained using standard hyperparameters, or were they altered? You may refer to the recent article, Augmentation of decision tree model through hyper-parameters tuning for monitoring of cutting tool faults based on vibration signatures.
Model was trained by both matlab classification learner and also by PyTorch in python. I was testing both tools for this application, and Matlab has more power in sense of testing several models and their optimalization at the same time. On the other hand, PyTorch has possibility to use voting classifiers, which can work better in specific situation. If this research would be for company, I would definitely build it using Python, but Matlab is much faster and easy to use. I used the optimized Ensemble bag tree classifier with altered (optimized) hyperparameters. It shows increase by 3 percent in accuracy in comparison without optimization.
Discuss the effect of kernel functions and regularization factor ‘C’ of SVM while training your data.
I had wrong title in Confusion matrix, at first tests I was using Support vector machine, but later on I switch to random trees, because they have higher accuracy.
Hyperparameters of the algorithm must be summarised in tabular form.
I have put there a summarizing table of hyperparameters.
Comment on computational time and complexity in the training of the algorithm.
I have added the specs of used PC, computational time of feature extracting from signals and images, training times of all tested models. The complexity lies mainly in the feature extraction function, which picks the right parts of signals (and spectrums) and images. Function for signal features originate from almost 4 years of my PhD, and currently is still in development. I plan to publish the function itself, because we use it already in several application (brick frost durability prediction, concrete thermal degradation classification, cavern localization), but at the current moment it is still not ready to be used by other researchers without any tweaks.
Was the data normalized/ standardized?
Data were not standardized or normalized. I have tried to apply Principal component analysis, but given the fact that image features helps distinguish 1200°C set from others, but has high variation, the resulting accuracy was not satisfying enough (around 66%).
How to ensure the robustness of the model in a highly noisy environment?
We can bring artificial noise to the dataset and test model with it, and check at what level of added noise the model has still satisfying accuracy and what level of noise is critical. From the feature extraction point the noise can be filtered out by proper hardware and software setup. I have added a short discussion on this topic.
Additionally, results can be provided considering different holdout % and holdout validation approaches. You may refer to this article to understand the holdout validation approach. https://doi.org/10.1115/1.4051696
I have added discussion on hold-out method and compare it with used cross validation. Holdout method is good for bigger datasets, and there is problem with representation of each classes if there is less observation. In my paper I have 7 classes, 197 observations and 41 features, so Holdout method is not from my honest opinion suitable. I have tried it but the accuracy is around 78-81 %. If we had less classes, it would work just fine.
How to deal with the diversity between the data distributions of present and future moments? ML-based algorithms can only resolve the regression/classification issues within the same data distributions. What would be the key steps of generalizing to unknown moments in predicting various parameters?
Well this is main problem for all papers focused on classification concrete properties by testing test samples by acoustic methods. Closed body of a cube or beam has nodes, and frequency modes, but concrete structures behave differently. If we stay in testing concrete specimens, then for the model to be successful our dataset would need to have full array of strength from 10 up to 120 MPa of compressive strength. Also, concrete can have different colour and complexity, so there would good opportunity for deep learning used on morphology and key aspects of different concrete mixtures. Other approach to it would be simulating the data input by the deep neural network, based on the done measurements. This can be view in the citation number xx, where volumetric morphology is simulated based on cross section images.
The authors do not mention the availability of this platform or framework for other practitioners, which is mandatory for this type of application. Add a comparison table at the end of the results section.
I have added the description of used model presets from Classification learner in Matlab, in the part Data availability statement there is link to table of extracted features with mixture and class labels, so anybody can download the table and test the exactly same model in Matlab or use different libraries in Python, C++ etc. The extraction function is planned to be published as well during 2023.
Once again, I thank you for your time and the review, I believe it make the paper more accurate and understandable.

Reviewer 2 Report
This is a paper with a novel theme. The content is quite rich. It is suggested to pay attention to some details and explain them, such as considering the influence caused by uneven lighting. It is suggested to add some descriptions of machine learning principles appropriately. In terms of references, it is recommended to pay attention to and refer to some relevant references, such as:
1. Diagnosis of internal cracks in concrete using vibro-acoustic modulation and machine learning
https://www.webofscience.com/wos/alldb/full-record/WOS:000844729800002
2. Identification and reconstruction of concrete mesostructure based on deep learning in artificial intelligence
https://www.webofscience.com/wos/alldb/full-record/WOS:000877344600003
3. Detection of source locations in RC columns using machine learning with acoustic emission data
https://www.webofscience.com/wos/alldb/full-record/WOS:000713265300002
Author Response
Reviewer 2:
This is a paper with a novel theme. The content is quite rich. It is suggested to pay attention to some details and explain them, such as considering the influence caused by uneven lighting. It is suggested to add some descriptions of machine learning principles appropriately. In terms of references, it is recommended to pay attention to and refer to some relevant references, such as:
Dear reviewer,
Thank you for your time and for your review. I have added the discussion of the different forms of illumination used in image processing. During testing we were tying different forms of postprocessing with adaptive brightness leveling, but it was not beneficial. I have added discussion to it.
For machine learning algorithm I have added description of the bagged trees, which is an ensemble method, with optimized hyperparameters (Tab 4) and a plot of generated decision tree. Unfortunately, the MATLAB function produce decision tree view with poor resolution, so I tried to compensate it by plotting it in vectors, so it doesn’t loose details when zoomed.
- Diagnosis of internal cracks in concrete using vibro-acoustic modulation and machine learning
I have added this reference with the discussion also about non-linear ultrasonic spectroscopy.
https://www.webofscience.com/wos/alldb/full-record/WOS:000844729800002
- Identification and reconstruction of concrete mesostructure based on deep learning in artificial intelligence
I have added the reference with the discussion of possible future development.
https://www.webofscience.com/wos/alldb/full-record/WOS:000877344600003
- Detection of source locations in RC columns using machine learning with acoustic emission data
I have added this paper with short discussion about detecting the most suitable parameters for ML and dimensionality reduction.
https://www.webofscience.com/wos/alldb/full-record/WOS:000713265300002
Once again, I thank you for your time and the review, I believe it make the paper more accurate and understandable.

Reviewer 3 Report
1. Abstract should be reorganized.
2. In the introduction of the paper, the problems or shortcomings of current method need to be added. How does the machine learning algorithms bridge this gap, and this point also should be addressed.
3. There are so many mixture parameters. How does identify them and if it would affect the accuracy of inputs for machine learning algorithms.
4. Lines 310 and 385, the figure numbers are missed.
5. The conclusions can be refined.
Author Response
Reiewer 3:
- Abstract should be reorganized.
Dear reviewer,
Thank you for your time and for your review. I have went thought he abstract a reorganize it and change some of the parts.
- In the introduction of the paper, the problems or shortcomings of current method need to be added. How does the machine learning algorithms bridge this gap, and this point also should be addressed.
I have added thorough discussion on ML topics, and why it is beneficial way of classification of this type of damage.
- There are so many mixture parameters. How does identify them and if it would affect the accuracy of inputs for machine learning algorithms.
The parameters of mixtures are very standard, and there is always change of only one parameter. The mixtures were designed by prof. Hela, and it is safe to say, that they represent the standard industry mixtures without any significant difference to what is being commonly used. The properties were set to observe the change of coarse aggregate and change of used binder, which are two factors which are most important in fire resistance of concrete. As can be seen on Fig. 8, the reference black markers are representing the range of resulting tensile strength, where the mixture ABC was with different coarse aggregate and DE was with different binder. The resulting span is beneficial for the creating of dataset of thermally degraded concrete, because there is higher chance that possible unknow specimen will lie in this range. With higher thermal degradation, the span of markers is getting lower, until it reaches 1000 °C where there is almost no difference among the mixtures. From this point of view it doesn’t depend on initial properties, because thermal degradation causes a normalization of physical-mechanical properties within the structure. After the partial sintering at 1200 the differences again occur. The biggest influence on input data would have curing of the specimens, if they were or were not pre-dried and so on. But this influence is highest at low temperatures (up to 200), where the concrete doesn’t loose so much of its mechanical properties.
- Lines 310 and 385, the figure numbers are missed.
I have check it and fixed them.
- The conclusions can be refined.
The conclusion was refined according to all of the reviews.
Once again, I thank you for your time and the review, I believe it make the paper more accurate and understandable.

Round 2
Reviewer 1 Report
That is a wonderful justification for all my comments.
According to your justification, your original title, 'Classification of thermally degraded concrete by acoustic resonance method and image analysis via machine learning,' seems useful and kept as it is.
Figure 17. Decision tree of optimized bagged tree model needs to be visible. The font size is too small. Could you revise it in the final version?
All other comments are addressed positively. Congrats.